# Influence of Scale Effect of Canopy Projection on Understory Microclimate in Three Subtropical Urban Broad-Leaved Forests

**Xueyan Gao** [1,2,3,4,†], **Chong Li** [1,2,3,4,†], **Yue Cai** [1,2,3,4], **Lei Ye** [1,2,3,4], **Longdong Xiao** [1,2,3,4], **Guomo Zhou** [1,2,3,4,*] **and Yufeng Zhou** [1,2,3,4]

1   State Key Laboratory of Subtropical Silviculture, Zhejiang A&F University, Lin'an 311300, China;
    gxy5225@stu.zafu.edu.cn (X.G.); chongli@zafu.edu.cn (C.L.); caiy@stu.zafu.edu.cn (Y.C.);
    ylei@stu.zafu.edu.cn (L.Y.); xiaold@stu.zafu.edu.cn (L.X.); zhouyf@zafu.edu.cn (Y.Z.)
2   Zhejiang Provincial Collaborative Innovation Center for Bamboo Resources and High-Efficiency Utilization,
    Zhejiang A&F University, Lin'an 311300, China
3   Key Laboratory of Carbon Cycling in Forest Ecosystems and Carbon Sequestration of Zhejiang Province,
    Zhejiang A&F University, Lin'an 311300, China
4   School of Environmental and Resources Science, Zhejiang A&F University, Lin'an 311300, China
*   Correspondence: zhougm@zafu.edu.cn
†   These authors contributed equally to this work.

**Abstract:** The canopy is the direct receiver and receptor of external environmental variations, and affects the microclimate and energy exchange between the understory and external environment. After autumn leaf fall, the canopy structure of different forests shows remarkable variation, causes changes in the microclimate and is essential for understory vegetation growth. Moreover, the microclimate is influenced by the scale effect of the canopy. However, the difference in influence between different forests remains unclear on a small scale. In this study, we aimed to analyze the influence of the scale effect of canopy projection on understory microclimate in three subtropical broad-leaved forests. Three urban forests: evergreen broad-leaved forest (EBF), deciduous broad-leaved forest (DBF), and mixed evergreen and deciduous broad-leaved forest (MBF) were selected for this study. Sensors for environmental monitoring were used to capture the microclimate data (temperature (T), relative humidity (RH), and light intensity (LI)) for each forest. Terrestrial laser scanning was employed to obtain the canopy projection intensity (CPI) at each sensor location. The results indicate that the influence range of canopy projection on the microclimate was different from stand to stand (5.5, 5, and 3 m). Moreover, there was a strong negative correlation between T and RH, and the time for T and LI to reach a significant correlation in different urban forests was different, as well as the time for RH and LI during the day. Finally, the correlation between CPI and the microclimate showed that canopy projection had the greatest effect on T and RH in MBF, followed by DBF and EBF. In conclusion, our findings confirm that canopy projection can significantly affect understory microclimate. This study provides a reference for the conservation of environmentally sensitive organisms for urban forest management.

**Keywords:** canopy projection; microclimate; canopy structural characteristics; scale effect; broad-leaved forest

## 1. Introduction

In recent years, there have been increasing concerns about the loss of biodiversity and sustainability of ecosystems due to global warming [1,2]. Global warming has considerably affected the growth processes of many plants [3,4]. A study showed that an increase in temperature by approximately 1 °C over several decades could result in changes in the species composition of forests [5]. In urban forest ecosystems, the forest canopy is the most direct receiver and receptor of external environmental changes, affecting the microclimate in forests by blocking solar radiation [6–8], which directly or indirectly

affects the growth of the understory and the material and energy cycling process of the ecosystem [9]. By comparing the microclimate in forests and the open areas nearby, it has been shown that tree canopies can regulate most of the meteorological variables, such as temperature and humidity [10–12], which is beneficial for climate-sensitive organisms (e.g., buckthorn) [13–15]. In addition to climate change, understanding the influence of canopy structure on the understory microclimate is crucial for biodiversity conservation.

Broad-leaved forests have been widely planted in southern Chinese cities because of their greening role and ecological values, such as water and soil conservation as well as air purification. The canopy structures of different broad-leaved forests significantly vary after autumn leaf fall, thereby leading to differences in their ability to block solar radiation penetration [16], which affects the understory microclimate. Canopy structural characteristics have been widely studied to explain the microclimate of plant communities. Hardwick et al. [17] reported that the forest microclimate varied less at sites with a higher leaf area index. Kong et al. [18] found that the volume of a three-dimensional point cloud of the leaves and shade better reflected the temperature variation in the forest compared to the leaf area index (LAI) and sky view factor. Jung et al. [19] concluded that there was a negative relationship between the canopy area and land surface temperature. Andrade et al. [20] reported that understory composition had a close relationship with canopy closure three decades after a forest fire. Kašpar et al. [21] found that canopy thickness and cover amplified temperature differences between understories and open areas. Owing to the heterogeneity of the spatial distribution of canopy structure, certain canopy parameters (e.g., canopy thickness, canopy closure, canopy cover area) are difficult to obtain directly. At present, indirect measurement methods are mostly used to obtain canopy structural parameters. Unmanned aerial vehicles (UAV) data were applied to derive tree heights and crown borders at the single-wood scale [22]. UAV-borne lidar and hyperspectral data were combined to acquire canopy height, LAI, and understory LAI [23].

Terrestrial laser scanning (TLS) has been widely used to determine forest structure characteristics [24] owing to its high accuracy, high efficiency, and lack of constraints on time and weather. Previous studies mostly used canopy parameters (such as leaf area index or canopy closure) to study their influence on the understory microclimate. Kong et al. [18] proposed a new method of using three-dimensional point cloud data that could reflect the vertical distribution of the canopy to study the influence of canopy on understory temperature.

Shading is an important driver affecting the microclimate [25]. Tree crowns, which are composed of branches, leaves, and twigs, can provide shade, and the shading effect depends on the shape and density of the canopy [26]. The shade quality is also affected by the trunks and leaves [27]. A study on hot-arid climatic regions identified that large trees could provide approximately 70% shade during spring and autumn [28]. Previous studies on shading have focused on the cooling effect of trees; for example, Akbari et al. [29] and Berry et al. [30] concluded that tree shade can lower residential surface temperatures by approximately 11–25 °C and 9 °C, respectively As the solar azimuth angle and solar altitude angle vary throughout the day, the canopy blocks solar radiation at different angles, thereby causing different levels of shading. However, research on the effect of shading changes on the microclimate in different forests during the day is limited. Some researchers have used the sky view factor and mathematical models to quantify tree shade [31,32]; nonetheless, because of the anisotropy of tree structures and the change in sun angle, the shading effect of the canopy was not completely simulated in these studies.

Microclimate characteristics are essential for the habitat requirements of climate-sensitive organisms [13]. In the past, most climate data were derived or modelled from weather stations [33]; therefore, the conclusions obtained by past studies were based on large-scale measurements [34]. However, the data obtained in this way considerably differ from the actual microclimate around the organisms [35]. The advent of sensors has bridged this gap and made the acquisition of microclimate data easier and faster. Nevertheless, there have only been a few studies on the influence of canopy projection on microclimate

at a small scale. For example, Rahman et al. [36] concluded that the daytime temperature of *Tilia cordata trees* decreased up to 3.5 °C within a radius of 4.5 m in August.

The objectives of this study were: (1) to explore the scale effect of canopy projection; (2) to determine the differences in correlation between microclimate factors of different forests; and (3) to reveal the regulatory effect of canopy projection on understory microclimate during daytime. TLS was applied to obtain canopy projection, and microclimate factors were measured using sensors for environmental monitoring on calm sunny autumn days. Correlation analyses were conducted to reveal the influence of the scale effect of canopy projection on microclimate during the day. The information obtained from this study can provide scientific data for urban forest development, biodiversity conservation, and forest fire prevention in this region, as well as serve as a reference for a better understanding of southern broad-leaved forest ecosystems.

## 2. Materials and Methods

### 2.1. Study Area

The study was conducted in Lin'an City (118°51′–119°52′E, 29°56′–30°23′N), Hangzhou, Zhejiang Province, China (Figure 1). The area has a subtropical monsoon climate, with an annual mean temperature of 16.4 °C, and the extreme maximum and minimum temperatures are 39 °C and −5 °C, respectively. The annual precipitation is 1628.6 mm [37], which is primarily observed during the flood season from May to September, with maximum precipitation in June. Affected by typhoons, hailstorms, and other extreme weather, the annual average frost-free period is 235 d [38], accounting for approximately 64% of the annual days. The landform is mainly composed of low hills and mountains [39]. The forestry area of Lin'an is 3,998,900 mu, and the forest coverage rate reaches 81.93% [40].

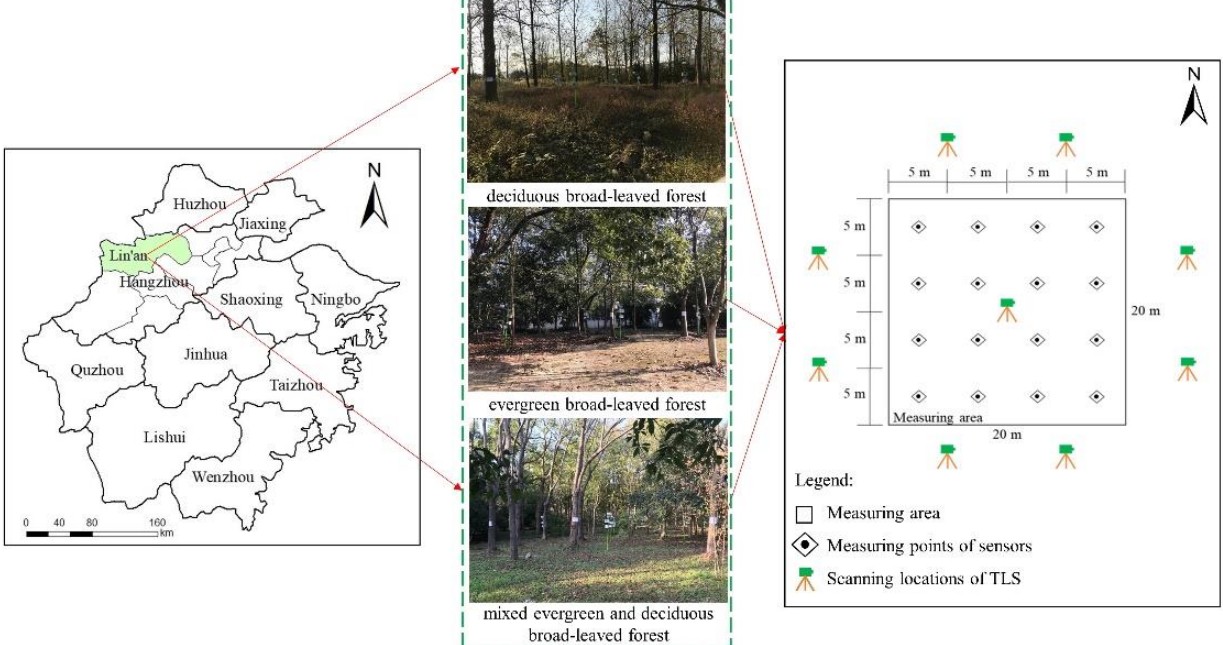

**Figure 1.** Study area and location of measuring points where microclimate factors, scanning locations of TLS, and canopy structural parameters were measured.

Three typical broad-leaved forests were selected for field observations at Zhejiang Agriculture and Forestry University (119°43′–119°44′E, 30°15′–30°16′N) in Lin'an district: evergreen broad-leaved forest (EBF), deciduous broad-leaved forest (DBF), and mixed evergreen deciduous broad-leaved forest (MBF). The main arbor species in these forests are *Cinnamomum camphora, Osmanthus fragrans*, and *Castanopsis sclerophylla* in EBF; *Populus simonii* and *Yulania liliiflora* in DBF; and *Koelreuteria paniculata, C. camphora, O. fragrans*, and

*Yulania liliiflora* in MBF. The standing conditions of these species were consistent, with good growth conditions and without any obvious influence of diseases and insect pests. More details on each forest are shown in Table 1. The measured area of the broad-leaved forests was 20 m × 20 m. Considering the influence of slope and edge effects, we chose sample plots located at a gentle slope and away from the road.

**Table 1.** Characteristics of the observation sites.

| Characteristics | EBF | DBF | MBF |
|---|---|---|---|
| Tree species | *Cinnamomum camphora, Osmanthus fragrans, Castanopsis sclerophylla* | *Populus simonii, Yulania liliiflora* | *Koelreuteria paniculata, Cinnamomum camphora, Osmanthus fragrans, Yulania liliiflora* |
| Shrub species | *Ilex chinensis Sims, Symplocos sumuntia* | *Broussonetia papyrifera* | *Pittosporum tobira* |
| Herbaceous plant species | *Reineckea carnea* | *Gynostemma pentaphyllum, Pennisetum alopecuroides* | *Imperata cylindrica* |
| Numbers | 28 | 8 | 33 |
| Average DBH/cm | 16.2 | 26.95 | 15.5 |
| Average TH/m | 10.9 | 16.4 | 13.2 |
| CCA/m$^2$ | 357.28 | 201.22 | 364.39 |
| CT/m | 8.51 | 14.3 | 11.07 |
| CV/m$^3$ | 1.13 | 1.06 | 1.58 |
| CC | 0.79 | 0.32 | 0.75 |
| LAI | 2.67 | 0.32 | 2.4 |

Note: DBH: diameter at breast height; TH: tree height; CCA: canopy cover area; CT: canopy thickness; CV: canopy volume; CC: canopy closure; LAI: leaf area index.

### 2.2. Temperature, Relative Humidity, and Light Intensity Data

Continuous sunny days with a stable weather from 6 November 2020 to 8 November 2020 were selected for microclimate measurements between 08:00 and 16:00. The time of sunrise and sunset was 06:16 and 17:08, respectively, during this period.

Temperature (T), relative humidity (RH), and light intensity (LI) were collected by high-accuracy sensors for environmental monitoring (Table 2), which were located at a height of 1.5 m. The data were recorded every minute and stored on secure digital (SD) cards. Considering that the differences between the different sensors could lead to inaccurate measurement results, the sensors were calibrated in terms of time and readings before use (Figure 2). Sixteen sensors were deployed in each plot, with a distance of 5 m between every two sensors, using a network point method (Figure 1) and monitored continuously for 3 d.

**Table 2.** Main parameters of sensor for environmental monitoring.

| Parameters | Temperature | Relative Humidity | Light Intensity |
|---|---|---|---|
| Range | −40–80 °C | 0–100% | 0–200,000 Lux |
| Accuracy | 0.1 °C | 0.3% | ±5% (25 °C) |
| Resolution | 0.1 °C | 0.1% | 1 Lux |

Processing of T, RH, LI Data

Microclimate factors (T, RH, LI) were processed as follows: first, because clouds and shadows of birds affect the measurement results, the mutation data were deleted and treated as outliers; second, T, RH, LI data were calculated by averaging the values over a 10-min period for 1 h; and third, based on the previous step, the mean values obtained over the same time and position were determined as stable T, RH, LI data during the experiment; thus, T, RH, LI data from 08:00 to 16:00 were acquired at 16 locations in each plot, and the measured results were shown to be relatively stable.

There was a direct relationship between microclimate factors and the proximity of the sensor to the heat source, as stated in Tobler's first law of geography, 'Everything is related to everything else, but closer things are more relative than more distant things'. Hence, the distribution diagrams of T, RH, and LI were constructed using the inverse distance weight method based on microclimate data at 16 locations in each plot in ArcGIS 10.2 (ESRI, USA) (Figure 3).

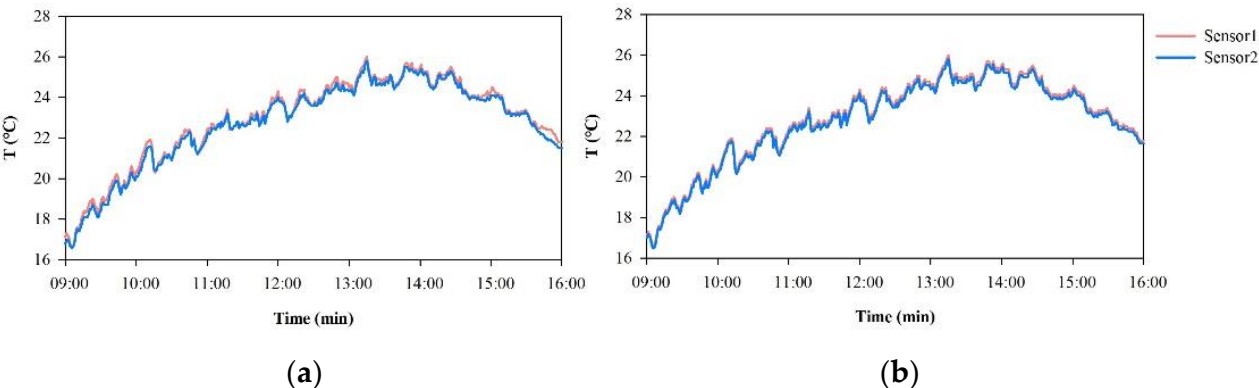

**Figure 2.** Least square linear fitting method was adopted to calibrate the temperature data of each sensor, with one sensor as the standard. The same method was used for relative humidity and light intensity. (**a**) Temperature (T) measurement data of sensors at the same place, (**b**) T data of sensors after calibration.

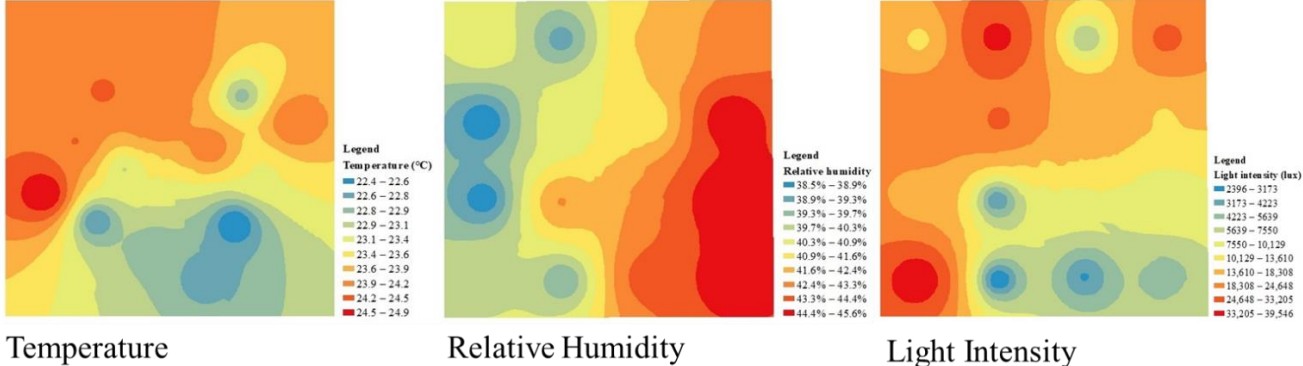

**Figure 3.** Interpolated graphs of microclimate data in mixed evergreen deciduous broad-leaved forest at 12:00. All results were divided into 10 levels. A deeper blue color indicates the lowest value, whereas a deeper red color indicates the highest value.

### 2.3. Point Cloud Data

Point cloud data refers to the collection of many points with three-dimensional coordinates (X, Y, Z) obtained via multi-angle scanning of objects using a TLS device (Leica ScanStation C05) (Table 3), which results in a large amount of data, hence the name 'point cloud' [41]. The TLS device has a scanning rotation angle of 0–360° in the horizontal direction and 0–270° in the vertical direction, and can scan in multiple directions. The errors of canopy occlusion and canopy point cloud underestimation can be minimized using the multi-scan approach (nine scans). During scanning, each station needed to be aligned with three targets at the sample site for point cloud registration in Cyclone (Leica Geosystems HDS, San Ramon, CA, USA).

**Table 3.** Characteristics of the Leica ScanStation C05 system.

| Parameters | Value |
| --- | --- |
| Field-of-view | 360° × 270° |
| Scan rate | 25,000 pts/s |
| Range | 35 m @ ≥ 5% albedo |
| Accuracy of position | 6 mm |
| Accuracy of distance | 4 mm |
| Spot size | 4.5 mm (FWHH-based); 7 mm (Gaussian-based) |
| Minimum point spacing | <1 mm |
| Operating temperature | 0–40 °C |

Cyclone is a software specialized for processing point cloud data. Before use, point cloud data were subjected to pre-processes, including registration, mergence, and denoising. Next, the data were processed to obtain structural parameters, such as diameter at breast height (DBH), canopy cover area (CCA) [42], tree height (TH) [43], canopy thickness (CT), canopy volume (CV) [18], and canopy projection intensity (CPI).

### 2.3.1. DBH, TH, CT, and CCA Extraction

The average DBH of the stand was obtained by calculating the mean values of all trees with transverse and longitudinal diameters at 1.3 m in forests (Figure 4a1,a2) using the point cloud slice-based method [42]. The distance from the tree base to the crown top is the TH of a single tree (Figure 4b), and the distance from the first branch point of the tree to the top of the crown is the CT of a single tree (Figure 4c). The average TH of all trees in a stand was the average TH of the stand, and the CT was calculated in the same way. Under the orthographic view of Cyclone, the CCA was calculated as the difference between the total plot area and the canopy gap area (Figure 4d).

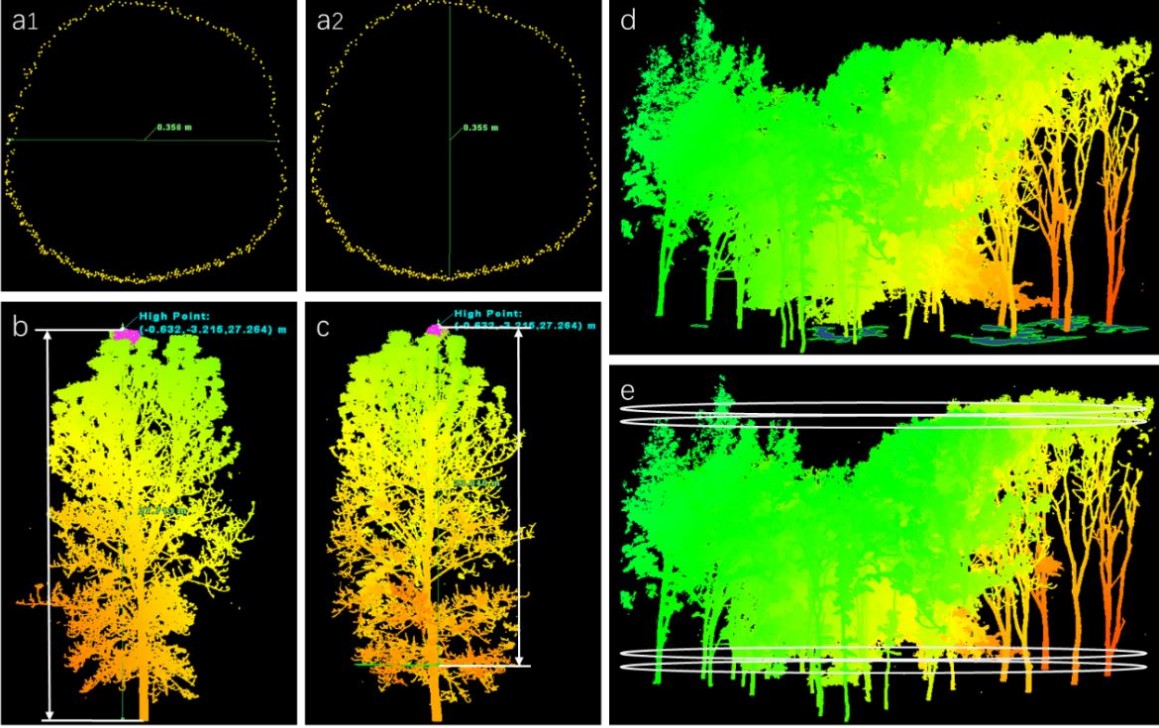

**Figure 4.** Extraction process of tree characteristics. (**a1**) Measurement of transverse diameter. (**a2**) Measurement of longitudinal diameter. (**b**) Measurement of tree height. (**c**) Measurement of canopy thickness. (**d**) Measurement of canopy cover area. (**e**) Vertical slices with 1 mm intervals were used to calculate canopy volume.

### 2.3.2. CV Extraction

Kong proposed to reduce the errors caused by crown gaps using a layered slice method to calculate the CV [18]. However, setting the size of each point cloud to 4 mm³ is assumed and not scientific. Hence, a pre-experiment was conducted. We selected a target with a known area as a scanning object, and a TLS device was placed at 5, 10, 15, 20, and 25 m because the height of most trees in the measured forest was between 5 and 25 m. The number of point clouds hitting the target was counted, and the area represented by each point cloud, which was the ratio of the target area to the number of point clouds on the target, was obtained (Figure 5). Next, the CV of the plot was calculated by programming with C# in Microsoft Visual Studio (Microsoft, USA). The process could be formulated as follows: first, point cloud data were sliced at 1 mm intervals from the bottom to the top of the crown along the Z-axis (Figure 4e); second, the point cloud area of each slice was summed (Equations (1)–(3)); third, the CV of each slice was calculated using Equation (4); and fourth, the total CV was summed, as shown in Equation (5).

$$d_i = \sqrt[2]{(x_i{}^2 + y_i{}^2 + z_i{}^2)} \tag{1}$$

$$S_i = 0.629 d_i{}^{2.218} \tag{2}$$

$$S_{hi} = \sum_{i=H_0}^{H} S_i \tag{3}$$

$$V_{hi} = 1 \times S_{hi} \tag{4}$$

$$V = \sum_{i=H_0}^{H} V_{hi} \tag{5}$$

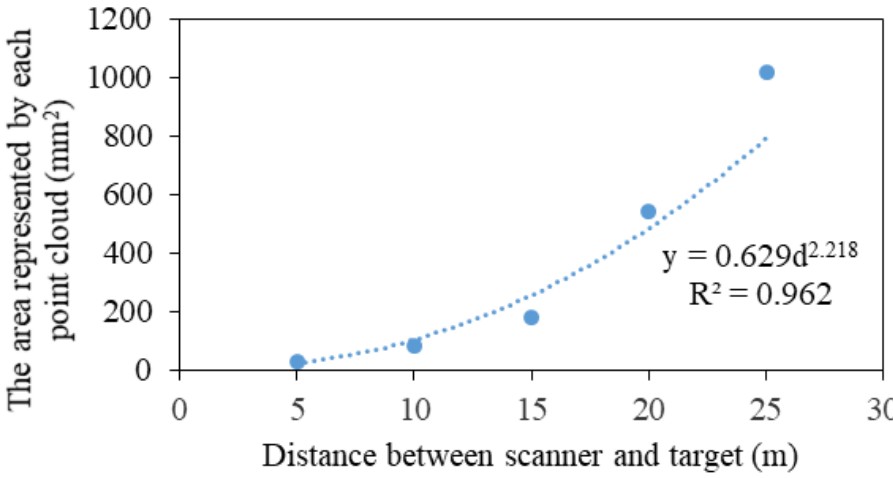

**Figure 5.** Relationship between the area represented by each point cloud and the distance from the scanner to the target.

In the above equations, $x_i$, $y_i$, and $z_i$ are the original coordinates of each point cloud; $d_i$ is the distance between the TLS and the point cloud; $S_i$ is the area represented by each point cloud; $S_{hi}$ is the total area of the point cloud of slice $i$; $H_0$ is the height of the bottom of the crown; $H$ is the height of the top of the crown; $V_{hi}$ is the CV of slice $i$; $V$ is the total CV.

### 2.3.3. CPI Extraction

The hourly variation in the canopy shading can be simulated using the known solar azimuth and altitude angles at different times of the day. In this study, solar azimuth and altitude angles were obtained by programming. The CPI is defined as the sum of the projected area of canopy point cloud per unit area, which reflects both the horizontal

and vertical distribution of the canopy through the superposition of shading. The specific process of CPI extraction is as follows (Figure 6):

1. The coordinates (*X*, *Y*, and *Z*) of each point were determined by the position of a TLS device. For the convenience of calculation, we obtained the azimuth angle of the forests through the electronic total station, and then the current coordinate system was converted to the geodetic coordinate system with the center of the sample plot as the origin.

2. The point cloud data of the canopy projection in the forest were generated from 8:00 to 16:00 using the projection transformation formula (Equations (6)–(10)).

3. Because the surrounding trees also affected the sample plot, the point cloud data outside the plot were removed according to the boundary of the plot after the projection transformation to obtain point cloud data of the canopy projection at different times.

4. The attribute of each point has four fields ($x_1'$, $y_1'$, $z_1'$, $S_i$), and the radius represented by each point was obtained according to the formula of area and radius (Equation (11)). The CPI was calculated by superimposing the buffers.

5. To explore which buffer provided the most relevant information on T and RH, we compared 20 different scales ranging from 0.5 m to 10 m, with an interval of 0.5 m at 12:00, when the solar altitude angle was the largest. The radius of the sensor probe (13 mm) was used as the buffer radius to describe the effects of canopy projection on LI, which was attributed to the fact that the light sensor measures the value of the current position.

6. Based on step five, the correlations between CPI and the microclimate factors in each forest were calculated from 08:00 to 16:00.

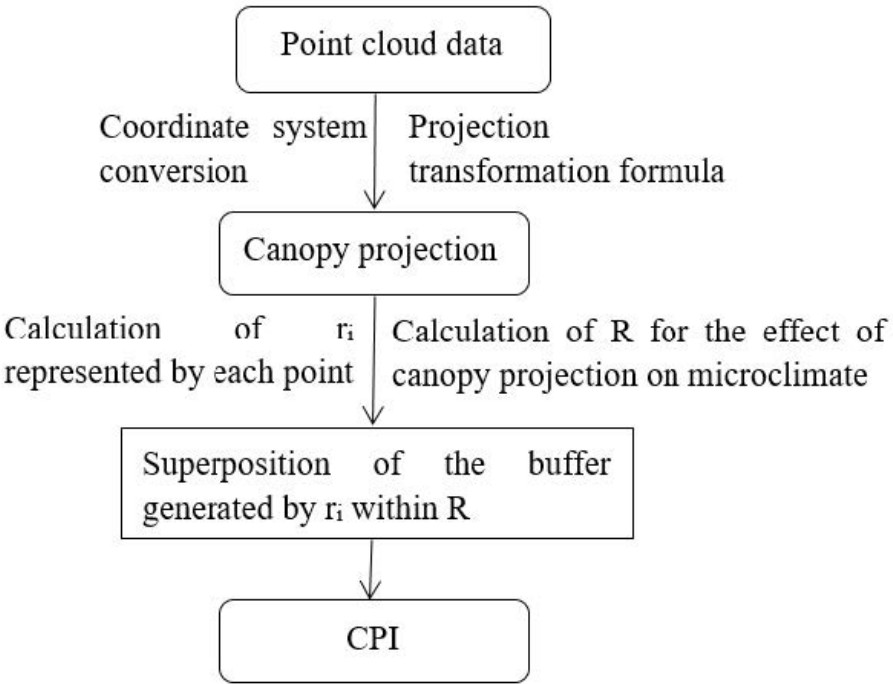

**Figure 6.** The process of CPI computation.

$$tan\partial = \frac{y_1' - y'}{x' - x_1'} \quad \partial \in (0, 90°) \tag{6}$$

$$tan(180° - \partial) = \frac{y_1' - y'}{x_1' - x'} \quad \partial \in (90, 180°) \tag{7}$$



$$tan(\partial - 180°) = \frac{y' - y_1'}{x_1' - x'} \quad \partial \in (180, 270°) \tag{8}$$

$$tan(360° - \partial) = \frac{y' - y_1'}{x' - x_1'} \quad \partial \in (270, 360°) \tag{9}$$

$$(x' - x_1')^2 + (y' - y_1')^2 = \left(\frac{z_1' - z'}{tan\beta}\right)^2 \tag{10}$$

$$r_i = \sqrt[2]{\frac{S_i}{\pi}} \tag{11}$$

$$CC = 1 - CO \tag{12}$$

In the above equations, $x'$, $y'$, and $z'$ are the new coordinates of each point cloud after coordinate transformation; $x_1'$, $y_1'$, and $z_1'$ are the new coordinates after projection transformation; $\partial$ is the solar azimuth angle; $\beta$ is the solar altitude angle; $S_i$ is the area represented by each point cloud; $r_i$ is the radius of each point cloud; $CC$ is canopy closure; $CO$ is canopy openness.

### 2.4. Canopy Image Data

In each forest, WinScanopy2013a for Hemispherical Image Analysis Instruments was used to take pictures of the canopy and sky from the ground to obtain canopy images using the five-point sampling method. The sample plot number and image number were recorded simultaneously. Canopy images were usually taken between 6:00–8:00 and 16:00–18:00 to avoid the influence of direct sunlight.

Leaf Area Index (LAI) and CC Extraction

Canopy images were digitized using WinScanopy2013a-containing computer analysis software. The LAI(2000G)-LogCI method [44] was used to calculate the LAI by adjusting the threshold to separate canopy and sky images. Next, the mean value of the five LAI values was considered as the LAI of this plot. Canopy openness was also obtained using the software, and the CC was calculated using Equation (12).

### 2.5. Statistical Analyses

The SPSS 22.0 software (IBM, USA) was used to perform correlation analyses of the above data. Pearson correlation coefficient analysis was conducted to describe the relationship between CPI and understory microclimate (T, RH, and LI) in each forest to explore the influence of canopy projection on the hourly variation of microclimate in different urban forests. P values less than 0.05 were considered significant, and those less than 0.01 were considered highly significant.

## 3. Results

### 3.1. Canopy Structural Characteristics

The canopy structural characteristics of the three forests, as captured by the point cloud data, are provided in Table 1. The highest values of CC and LAI were observed in EBF, the highest CT value was in DBF, and the highest CCA and CV values were in MBF.

### 3.2. Canopy Projection Scales

The correlation results between CPI and microclimate factors at 20 canopy projection scales are shown in Figure 7. CPI showed a significant correlation to T and RH between 2.5–4 m in EBF, 3–10 m in DBF, and 0.5–10 m in MBF.

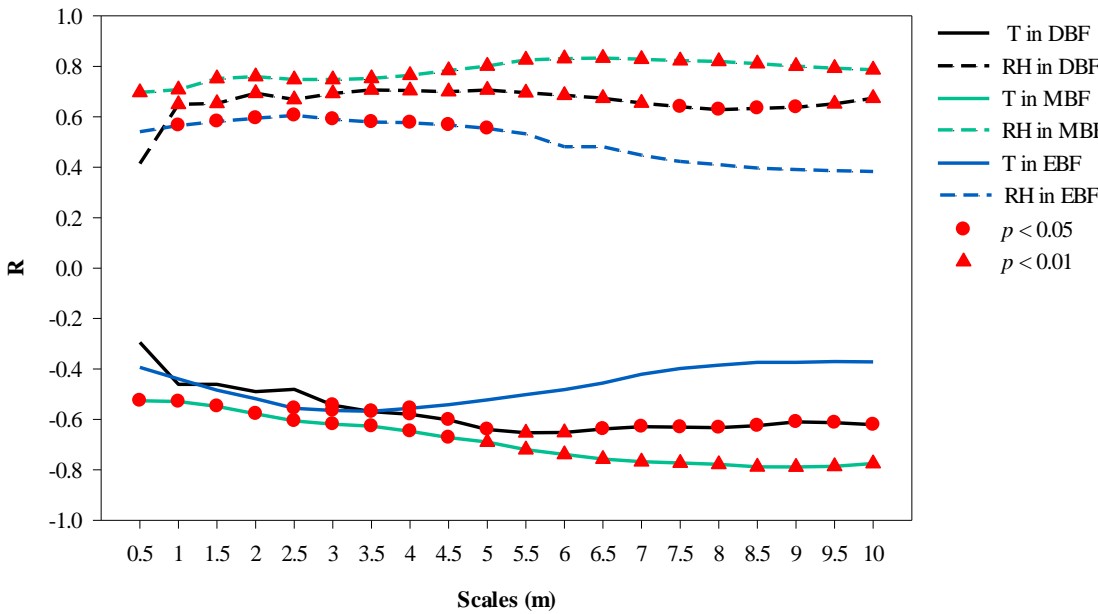

**Figure 7.** Correlation of canopy projection intensity to temperature (T) and relative humidity (RH) with scales at 12:00 in the three forests.

### 3.3. Hourly Variations in CPI and Microclimate Factors

#### 3.3.1. Hourly Variation in CPI

The average CPI was calculated hourly during the daytime, as shown in Figure 8. Statistical analysis indicated that there were significant differences in CPI between the three forest types. The highest CPI (6.29) was observed in MBF, which maintained higher values in the morning compared with that of the EBF, which had higher values in the afternoon. The CPI gradually increased until 12:00, when the peak value occurred in DBF.

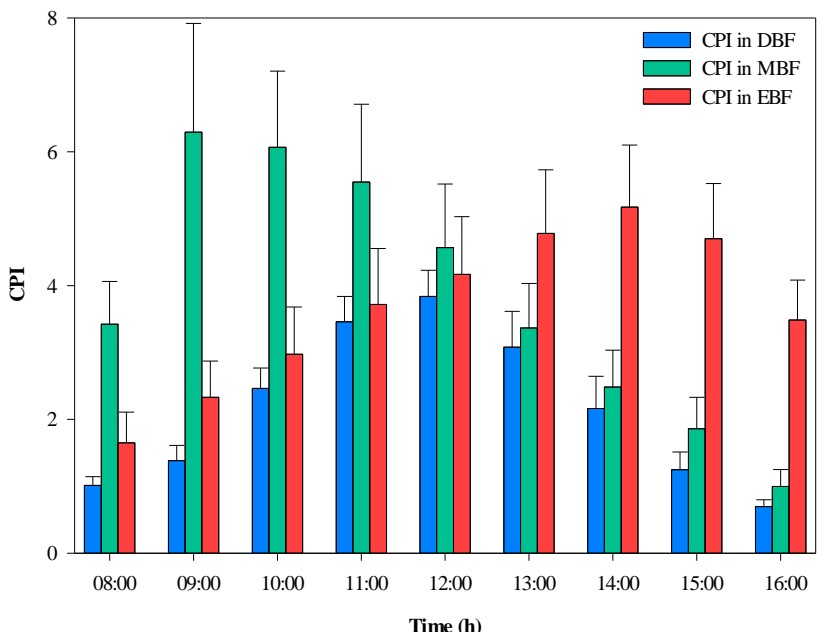

**Figure 8.** Diurnal variation in the average canopy projection intensity (CPI) within the three broad-leaved forests. EBF, evergreen broad-leaved forest; DBF, deciduous broad-leaved forest; MBF, mixed evergreen, and deciduous broad-leaved forest.

### 3.3.2. Hourly Variation in the Microclimate Factors

The variations in the microclimate were found to differ from forest to forest over a day. The greatest T (25.4 °C) was observed in DBF, followed by the MBF (24.5 °C) and EBF (23.8 °C) at 13:00 (Figure 9a). It was obvious that the T in DBF was higher than that in EBF and MBF, and that the difference was greatest between 13:00 and 14:00. The hourly variation in RH showed a U-shaped continuous change (Figure 9b), and the peaks occurred at 8:00. Moreover, the minimum occurred at 14:00 in EBF (43.85%), whereas the minimum occurred at 13:00 in DBF (37.14%) and MBF (38.96%). The difference in RH was most obvious between 12:00 and 13:00. Furthermore, the LI varied significantly depending on the type of forest (Figure 9c). It was clear that the LI of DBF was remarkably higher by almost 2–3 times than that of EBF and MBF between 11:00 and 13:00.

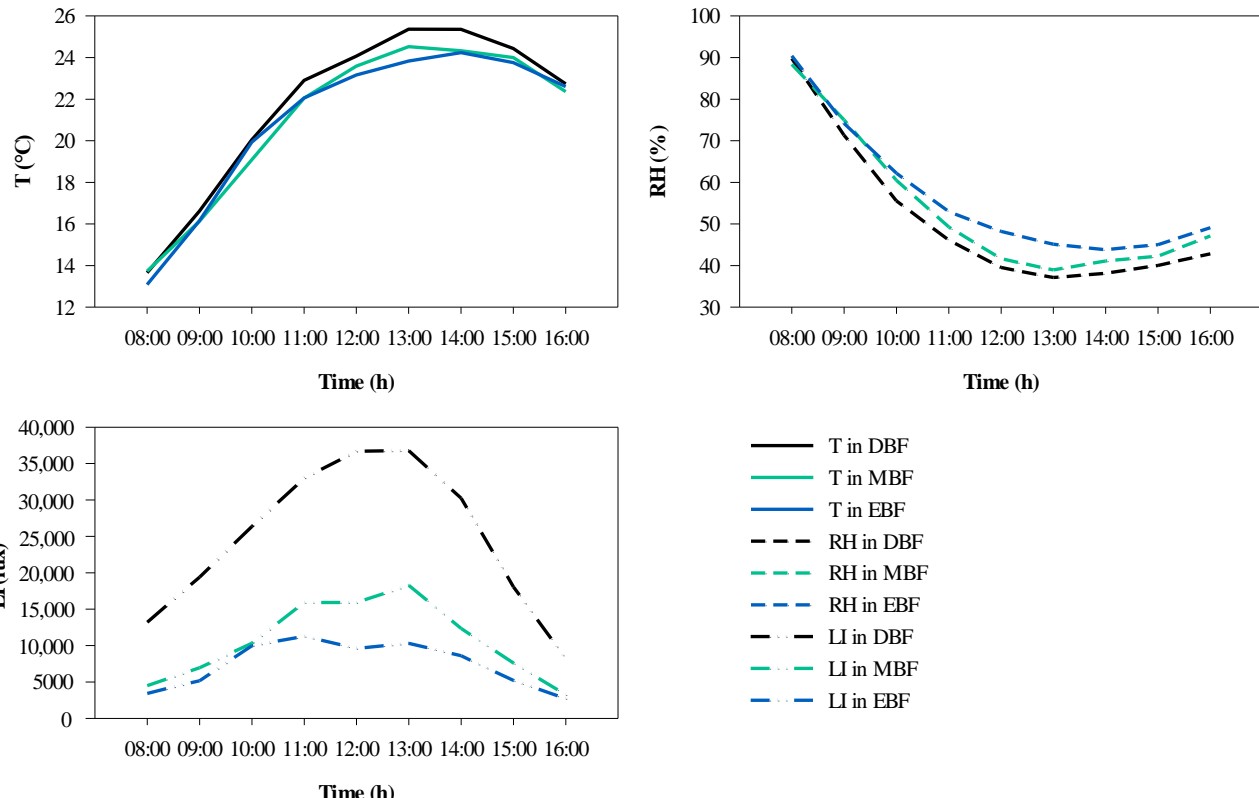

**Figure 9.** Curves of hourly variation in microclimate factors in the three forests over a day. (**a**) The average temperature (T), (**b**) average relative humidity (RH), and (**c**) average light intensity (LI).

### 3.4. Correlation Analyses between Microclimate Factors

Correlation analyses were performed to elucidate the mutual effects between the microclimate factors in the different broad-leaved forests at different times of the day in autumn, and the results are shown in Figure 10. Between 08:00 and 16:00, there was a significant negative correlation between T and RH (Figure 10a) in EBF (average r = −0.87, $p < 0.01$) and DBF (average r = −0.76, $p < 0.05$), and the correlation coefficients remained stable; in contrast, MBF showed a decreasing trend (average r = −0.72, $p < 0.05$). In addition, the correlation between T and LI (Figure 10b) was significant from 11:00 to 15:00 (except at 12:00) in EBF and significant from 09:00 to 13:00 in DBF, with the longest duration of significant correlation shown in MBF from 10:00 to 15:00. In terms of the correlation between RH and LI (Figure 10c), the time period when RH and LI were significantly correlated was 11:00–15:00 (except at 12:00) in EBF, 10:00–13:00 in DBF, and 11:00–14:00 in MBF.

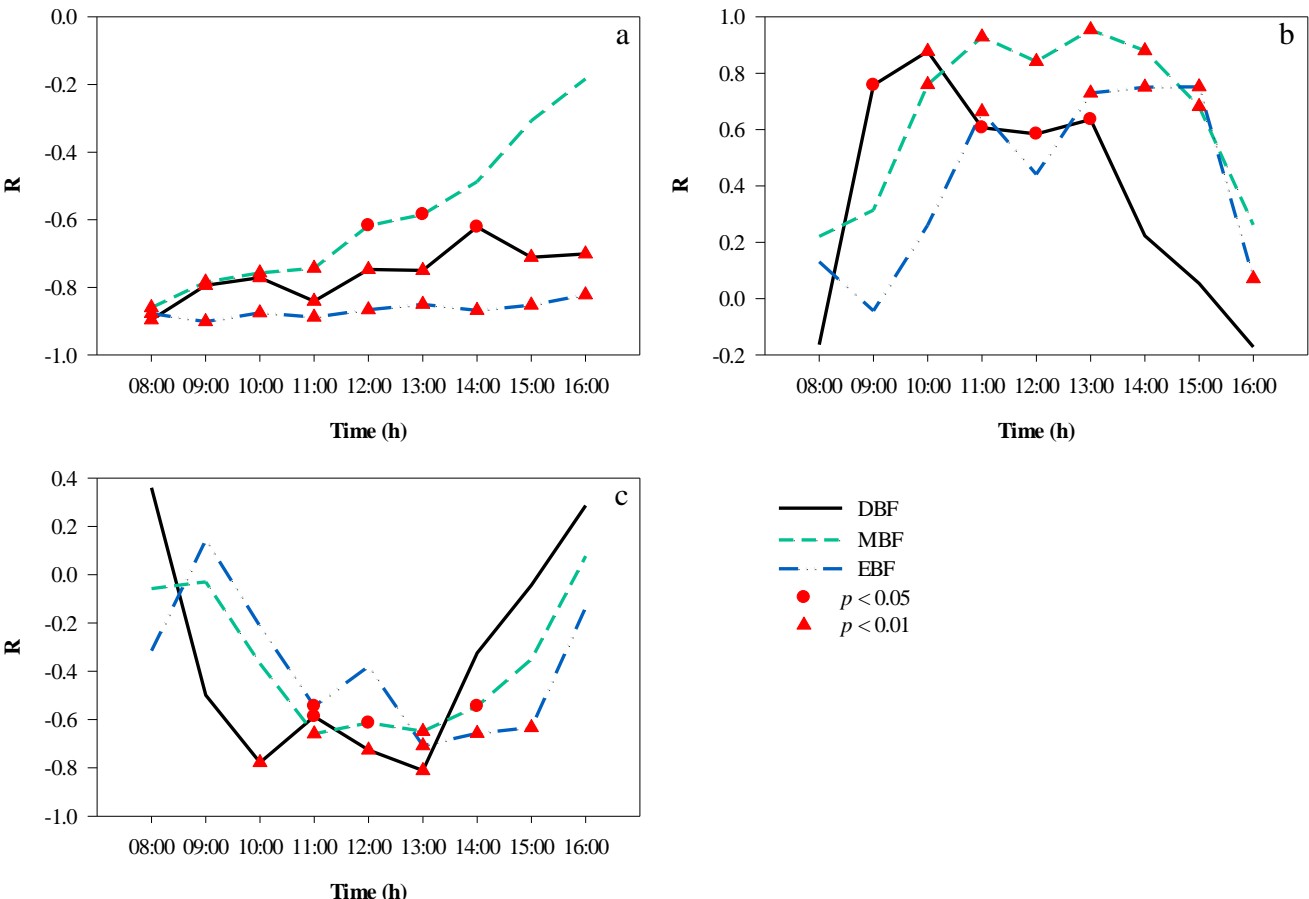

**Figure 10.** Hourly correlation among understory microclimate factors over a day. (**a**) T and RH, (**b**) T and LI, (**c**) RH and LI. LI, light intensity; RH, relative humidity; T, temperature.

It was clear that the correlation between RH and LI was the opposite of that between T and LI. We also found that the time required for T and LI to reach a significant correlation was 1 h earlier than that for RH and LI in DBF and MBF, but the times were synchronous in EBF. Moreover, the correlation between LI and T was greater than that between LI and RH at most times of the day.

### 3.5. Correlation Analyses between the CPI and Microclimate Factors

The correlation analyses between the CPI and T for the three forests are presented in Figure 11a. Between 08:00 and 16:00, the canopy projection had a changing impact on T. A significant negative correlation was found between 11:00 and 15:00 in EBF, between 08:00 and 13:00 in DBF, and between 10:00 and 15:00 in MBF. During these periods, the time to peak was also different: at 11:00 in EBF ($r = -0.58$, $p < 0.05$) and MBF ($r = -0.84$, $p < 0.01$), and at 10:00 in DBF ($r = -0.67$, $p < 0.01$).

CPI and RH showed a positive correlation between 08:00 and 16:00 (Figure 11b). The longest significant correlation was maintained in DBF (08:00–13:00) and MBF (10:00–15:00), followed by the EBF (11:00–15:00). In the morning, as RH decreased and CPI increased gradually, the coefficient between CPI and RH increased, peaking at 12:00 in DBF ($r = 0.70$, $p < 0.01$), MBF ($r = 0.802$, $p < 0.01$), and EBF ($r = 0.59$, $p < 0.05$). In the afternoon, the effect of canopy projection on the RH became weaker.

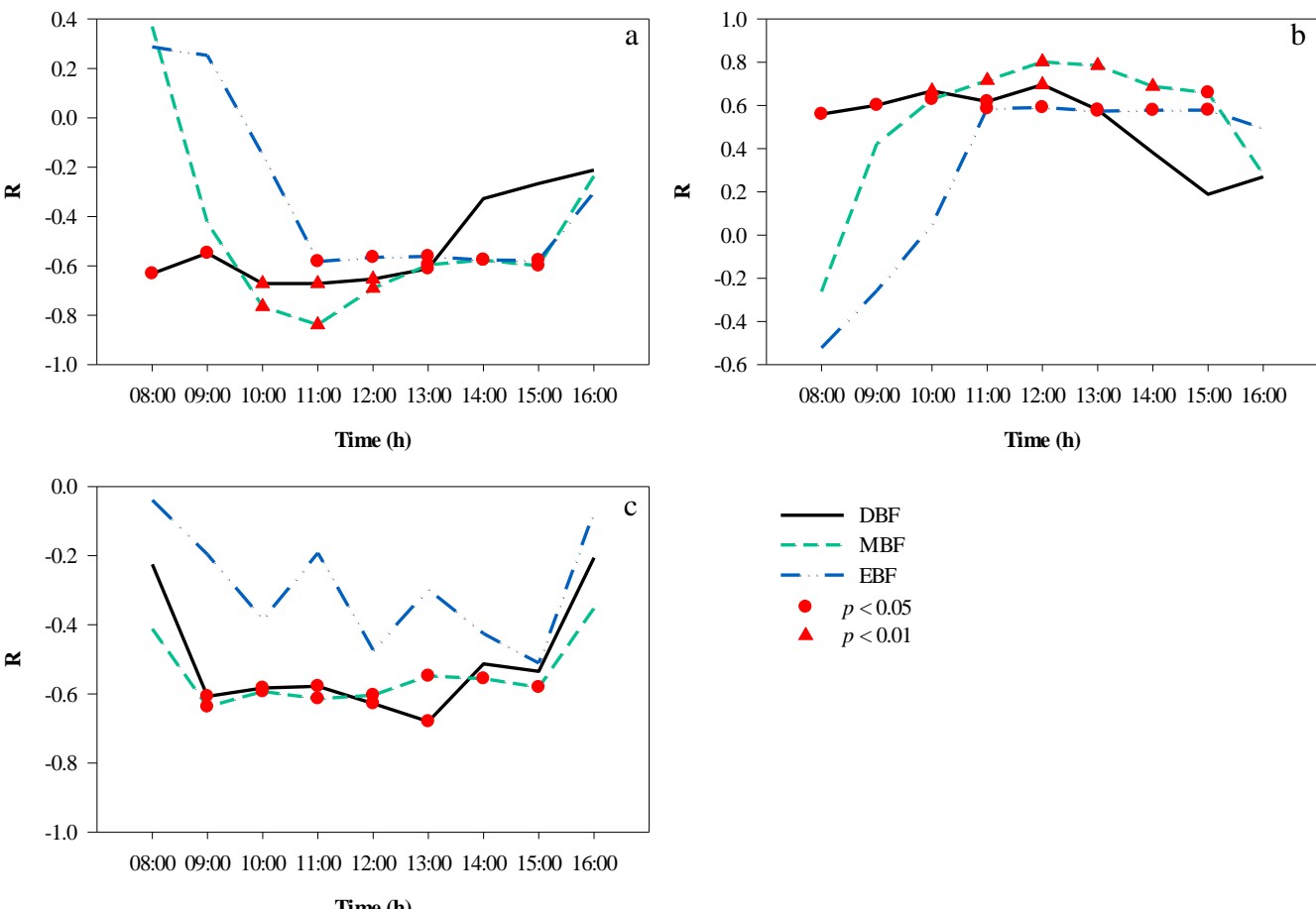

**Figure 11.** Hourly correlation between CPI and microclimate factors over a day. (**a**) CPI and T, (**b**) CPI and RH, (**c**) CPI and LI. CPI, canopy projection intensity; LI, light intensity; RH, relative humidity; T, temperature.

The hourly correlation coefficients between the CPI and LI are shown in Figure 11c. In EBF, there was no significant correlation between CPI and LI over a day, which indicated that canopy projection could affect LI to some extent. In MBF, there was a significant negative correlation between CPI and LI from 09:00 to 15:00, and the correlation coefficients fluctuated at approximately −0.60. In DBF, after 09:00, the correlation between the two variables strengthened and was significant until 13:00, when it reached its peak (r = −0.68, $p < 0.05$).

Overall, the absolute values of the correlation coefficients between CPI and RH were higher than those between CPI and T at most times of the day in the three forests.

Figure 12 shows the spatial distribution pattern of the effect of canopy projection on T from 08:00 to 16:00 on a scale of 20 m × 20 m. As displayed by the map, the denser the projection, the lower the T is, as indicated by blue. Conversely, the less dense the projection, the redder the color and the higher the T.

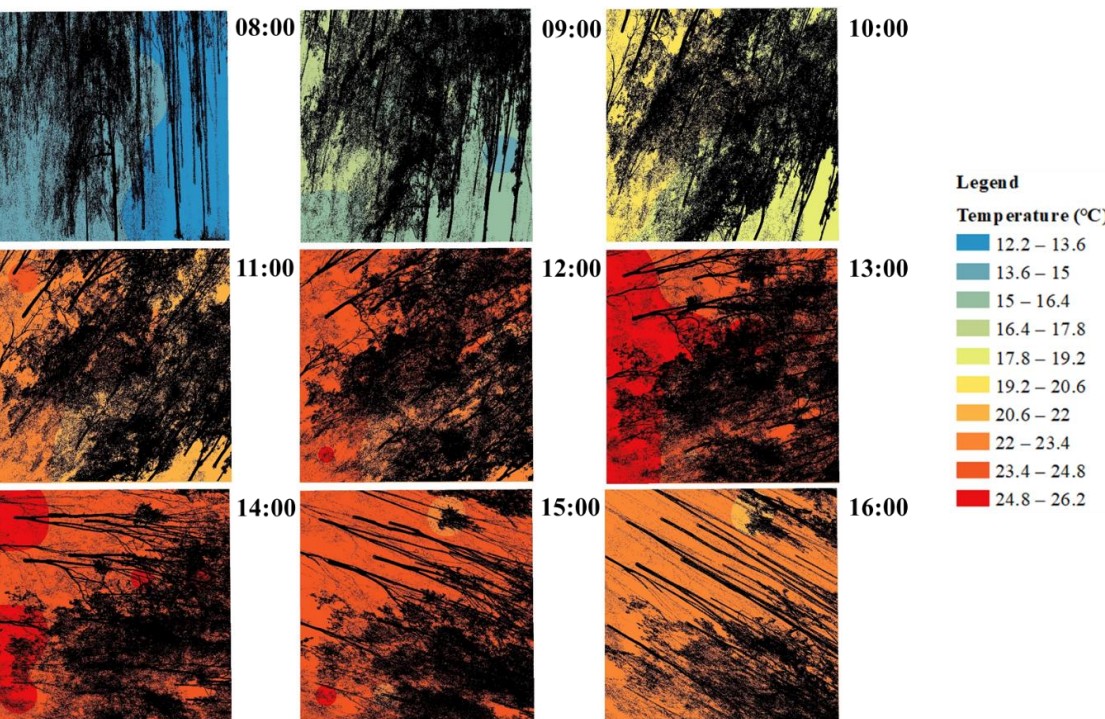

**Figure 12.** Interpolation results of temperature (°C) during the day in mixed evergreen and deciduous broad-leaved forest and hourly superposition effect of canopy projection.

## 4. Discussion

### 4.1. Scale Effect of Canopy Projection

Trees in urban forests provide light and space in different ways to regulate T and RH under the canopy, thus affecting the growth of the understory vegetation [11,45,46]. Nevertheless, existing studies have focused mainly on the effect of the canopy on the daily average variations in T and RH or on the data of a certain period, ignoring the more detailed variation during the day, which is important for studying the actual variation of the microclimate and the effect of canopy structural characteristics on the understory environment [47]. Therefore, we explored the hourly effects of canopy structural characteristics on the understory microclimate, and our findings confirmed that canopy projection significantly affected the microclimate in the forest ecosystems studied.

Most previous studies used canopy coverage to describe the effect of canopy projection on the microclimate, which varied with spatial scales and there were differences in the results of different regions. In Taiwan, the T of the measured point was influenced by vegetation coverage within a radius of 10 m around the measured point [48]. In the United States, when canopy coverage exceeded 40% within a radius of 60–90 m, the increased T of the impervious surface could be offset by the decreased T of trees [49]. In Brazil, the radius of the buffer zone affecting the T of the measurement point was 125 m [34]. However, these conclusions were mostly based on large-scale measurements. To date, only a few studies have been conducted to select different buffer radii for different forests, using small-scale plant communities (e.g., Rahman et al. [36]) as the starting point. In the present study, the correlation of CPI with T and RH at 12:00 was used to determine canopy projection scales, and 5, 5.5, and 3 m were selected as the projection scales in MBF, DBF, and EBF, respectively, where CPI had highly significant correlation with RH and T simultaneously ($p < 0.01$).

### 4.2. Correlation of Microclimate Factors in Different Urban Forests

It is well known that T and RH are negatively correlated in forests [50–52]. This relationship was confirmed in the present study. The correlations of LI with T and RH at different times of the day indicated that the time required for LI and T to reach a significant

correlation differed between forests, which was at 09:00 in DBF, 10:00 in MBF, and 11:00 in EBF. In addition, LI and T reached a significant correlation 1 h earlier than LI and RH in DBF and MBF. In contrast, in EBF, the times were synchronized. This could be explained by the fact that among the three forests, EBF had the largest CC and LAI as well as the largest number of leaves per unit area, causing water loss inside the forest to be slow and humidity to be relatively high at a certain period [53], which led to differences in the time for LI and T to reach significant correlations in the different forests. Nonetheless, the question as to which forest (DBF or MBF) first reached a significant correlation between LI and RH needs to be further explored by shortening the time scale.

This study also found that the correlation between LI and T was greater than that between LI and RH at most times of the day, which was because T in the forest is mainly influenced by solar radiation [46], whereas RH is related to solar radiation, plant transpiration [54,55], soil moisture content, and soil evaporation in the forest [56,57].

### 4.3. Effect of Canopy Projection on the Microclimate

The rotation of the Earth causes the solar altitude angle to vary throughout the day, and the degree of solar radiation blocked by a canopy varies in different forests, causing the correlation between canopy structure characteristics and microclimate to change throughout the day [53]. Microclimate variations in plant communities were obvious during the day [58], and the canopy had more significant effects on the microclimate factors during daytime hours; thus, the experiment was conducted between 08:00 and 16:00.

In our study, there was a negative but weak correlation between CPI and LI in EBF, which was attributed to the larger LAI of this forest and less LI entering into the forest [59] (Figure 9c). This led to a smaller correlation between CPI and LI. However, CPI showed a larger correlation with LI in both the DBF and MBF.

The results of the correlation analyses between CPI and T in the three broad-leaved forests confirmed the findings of previous studies [6,18,58,60], as they showed a significant negative correlation, further confirming the influence of diurnal variation in canopy projection on T in a forest. This study showed that the maximum correlation between CPI and T occurred in MBF, followed by DBF and EBF. One possible explanation is that MBF had the largest CCA, CV, and CPI, which maintained a lower T in the forest. Moreover, the correlation may be affected by the understory vegetation. Han [61] analyzed the characteristics of understory vegetation communities of five forests in Shanghai and found that the largest plant diversity index was in EBF, followed by coniferous forest, DBF, MBF, and bamboo forest. However, the range of T affected by the understory vegetation remains to be studied in our subsequent experiments.

The effect of the canopy on RH in forests has been widely studied [7,50,62,63], and the current study showed a positive correlation between canopy and RH in general. Our experiments indicated that the maximum correlation between CPI and RH was shown in MBF, which has a larger RH, followed by DBF and EBF. This result is similar to the maximum correlation between CPI and T. Similarly, the extent to which RH is affected by the understory vegetation needs to be explained in subsequent studies.

In addition, the time required for CPI and RH to reach the peak of correlation in the three forests (12:00) was later than that for CPI and T (10:00–11:00), which was mainly attributed to the fact that as CPI increased during the day, T decreased, whereas RH increased in the forest [50]. RH was also affected by the evapotranspiration of plant communities [64]. Consequently, CPI and RH peaked later than CPI and T, and the effect of CPI on RH was greater than that on T during the day (Figure 11), which is consistent with the finding of Porté et al. [65].

### 5. Conclusions

This study analyzed the influence of hourly variation in canopy projection on the microclimate factors and its scale effect. The findings are as follows: first, scales of canopy projection on the microclimate were 5 m in MBF, 3 m in EBF, and 5.5 m in DBF; second,

some common results were obtained for the broad-leaved forests. The correlation between LI and T was greater than that between LI and RH at most times of the day, and the time of CPI and RH to reach the peak of correlation (12:00) was later than that of CPI and T (10:00–11:00), given that T is mainly affected by LI in the forest, whereas RH is also affected by the evapotranspiration of plant communities. Third, in this study, results vary considerably between different forests. In both the DBF and MBF, the time for LI and T to reach a significant correlation was 1 h earlier than that for LI and RH, and the maximum correlation of CPI to T and RH occurred in MBF, followed by DBF and EBF. These findings confirm that canopy projection could significantly affect the microclimate, which is important for the conservation of climate-sensitive organisms and forest fire prevention in southern urban forests. However, this study had some limitations. The effect of understory vegetation on the microclimate was ignored, and plant transpiration will be considered in future studies.

**Author Contributions:** Funding acquisition, G.Z.; conceptualization, G.Z. and Y.Z.; methodology, X.G., C.L., G.Z. and Y.Z.; supervision, C.L. and Y.Z.; project administration, Y.Z.; investigation, X.G., C.L., L.Y. and L.X.; data curation, X.G.; visualization, X.G., Y.C. and C.L.; writing—original draft preparation, X.G.; writing—review and editing, Y.C., C.L. and Y.Z. All authors have read and agreed to the published version of the manuscript.

**Funding:** This research was funded by the National Natural Science Foundation of China (Grant number: U1809208), and the Key Research and Development Program of Zhejiang Province (Grant number: 2021C02005).

**Institutional Review Board Statement:** Not applicable.

**Informed Consent Statement:** Not applicable.

**Data Availability Statement:** Not applicable.

**Acknowledgments:** This study was supported by the National Natural Science Foundation of China (Grant number: U1809208), and the Key Research and Development Program of Zhejiang Province (Grant number: 2021C02005). We would also like to thank the editor and anonymous reviewers for their contributions to the peer review of our work.

**Conflicts of Interest:** The authors declare no conflict of interest.

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
