# Peer review of "Influence of Scale Effect of Canopy Projection on Understory Microclimate in Three Subtropical Urban Broad-Leaved Forests"

_remotesensing, doi:10.3390/rs13183786_

Round 1

Reviewer 1 Report

You can find my comments attached.

Good luck!

Reviewer 2 Report

The article presents a study about the influence of canopy projection on understory microclimate in three broad-leaved forests. The article is well distributed and easy to read and understand by the reader. The introduction is clear, and the methodology is well explained. 

A few questions: 
-    I’m not sure if only three sites could provide enough results because of the high heterogeny of forest sites. Could the authors discuss it? 
-    There are quite acronyms that could be confused for the reader. In this case, is possible to create an appendix with a list and definition of each one?
-    Please, follows botanical scientific nomenclature when named one forest species. 

Finally, this article has enough quality to been published in remote sensing
